# Cerebral White Matter Hyperintensity as a Healthcare Quotient

**DOI:** 10.3390/jcm8111823

**Published:** 2019-11-01

**Authors:** Kaechang Park, Kiyotaka Nemoto, Yoshinori Yamakawa, Fumio Yamashita, Keitaro Yoshida, Masashi Tamura, Atsushi Kawaguchi, Tetsuaki Arai, Makoto Sasaki

**Affiliations:** 1Research Organization for Regional Alliance, Kochi University of Technology, Kochi 782-0003, Japan; 2Department of Psychiatry, Faculty of Medicine, University of Tsukuba, Tsukuba 305-8575, Japan; kiyotaka@nemotos.net (K.N.); keicima0154@gmail.com (K.Y.); tamuramas@gmail.com (M.T.); 4632tetsu@md.tsukuba.ac.jp (T.A.); 3ImPACT Program of Council for Science, Technology and Innovation (Cabinet Office, Government of Japan), Tokyo 100-8974, Japan; yamakawa@bi-lab.org; 4Division of Ultrahigh Field MRI, Institute for Biomedical Sciences, Iwate Medical University, Morioka 028-3694, Japan; fyamashi@iwate-med.ac.jp (F.Y.); masasaki@iwate-med.ac.jp (M.S.); 5Center for Comprehensive Community Medicine, Faculty of Medicine, Saga University, Saga 849-8501, Japan; akawa@cc.saga-u.ac.jp

**Keywords:** white matter hyperintensity, MRI, healthcare quotient, chronic

## Abstract

To better understand the risk factors and optimal therapeutic strategies of cerebral white matter hyperintensity (WMH), we examined a large population of adults with and without various vascular risk factors (VRFs) or vascular risk conditions (VRCs), such as hypertension (HT), diabetes mellitus (DM), and dyslipidemia (DLP), including the comorbidities. We assessed two participant groups having no medical history of stroke or dementia that underwent brain checkup using magnetic resonance imaging (MRI): 5541 participants (2760 men, 2781 women) without VRCs and 1969 participants (1169 men, 800 women) who had received drug treatments for VRCs and the combination of comorbidities. For data analysis, we constructed WMH-brain healthcare quotient (WMH-BHQ) based on the percentile rank of WMH volume. This metric has an inverse relation to WMH. Multiple linear regression analysis of 5541 participants without VRCs revealed that age, systolic blood pressure (SBP), Brinkman index (BI), and female sex were significant factors lowering WMH-BHQ, whereas body mass index (BMI), male sex, fasting blood sugar, and triglyceride levels were increasing factors. The Kruskal–Wallis test and Dunn tests showed that WMH-BHQs significantly increased or decreased with BMI or SBP and with BI classification, respectively. Regarding the impact of impaired fasting glucose and abnormal lipid metabolism, there were almost no significant relationships. For 1969 participants who had HT, DM, and DLP, as well as their comorbidities, we found that DLP played a substantial role in increasing WMH-BHQ for some comorbidities, whereas the presence of HT and DM alone tended to decrease it. Cerebral WMH can be used as a healthcare quotient for quantitatively evaluating VRFs and VRCs and their comorbidities.

## 1. Introduction

Cerebral vessel diseases are classified as large vessels diseases (LVDs) or small vessels diseases (SVDs) based on whether the diameters of the vessels involved are larger than a few millimeters or smaller than several hundred micrometers, respectively [1,2]. Both categories can be noninvasively diagnosed using magnetic resonance imaging (MRI) [3]. With regard to risk factors, many longitudinal studies have reported that LVD can be responsible for stroke, cognitive decline, and dementia [1,2,3,4]. Hypertension (HT), diabetes mellitus (DM), and dyslipidemia (DLP), three primary vascular risk conditions (VRCs) in developed countries, are risk factors for LVD [5,6]. Although HT is an obvious risk factor for SVD, the roles of DM and DLP remain disputable [7,8]. Compared with LVD, SVD has not yet been sufficiently studied with regard to its onset and development. The pathological complexity of SVD, such as arteriosclerosis, hyalinosis, blood–brain barrier disruption, and venous collagenosis, have long complicated our ability to fully comprehend its many aspects [1,8]. For SVD studies using MRI, it is difficult to include large numbers of participants who have various conditions ranging from preclinical to chronic HT, DM, and DLP, partly because SVD is mostly asymptomatic and does not have hospital follow-up like LVD. To clarify the whole range of cerebral vessel damages, a large scale epidemiological study of SVD including preclinical or chronic HT, DM, and DLP is essential.

Brain MRIs show four major features of SVD: lacuna stroke, white matter hyperintensity (WMH), cerebral microbleeds, and visible perivascular spaces [9]. In our study, we focused on WMHs, also known as leukoaraiosis, which are commonly observed in the general population, particularly among individuals with preclinical or chronic HT, DM, and DLP. WMHs are recognized in >60% of people over 60 years old [10] and >30% of people with the age range from 40 to 50 years in Japan, where MRI examination is incorporated as part of health checkups in connection with a screening program called Brain Dock [11,12]. WMHs are regarded as disappearance of arterioles and capillary arteries caused by aging, HT, and reduced cerebral blood flow [9,13]. WMHs are also significantly associated with recurrent stroke, cognitive decline, and dementia [2,5].

Numerous efforts have been made to develop MRI-based measures of health status, such as the concept of “brain age”, which reportedly reflects the mortality of an individual [14]. Our team earlier proposed brain healthcare quotients (BHQs) based on gray matter volume or fractional anisotropy and found significant associations between these proffered metrics and various physical factors, such as obesity, high blood pressure (BP), and daily personal schedules, as well as social factors, including subjective socioeconomic status, subjective well-being, and the adoption of a postmaterialism view of life [15]. In this cross-sectional study, we proposed another BHQ based on WMH, which begins to appear in early middle age and increases in frequency with age. Using an extensively large database obtained from 8921 participants who were examined through MRI as part of the Brain Dock component of a routine health checkups, we analyzed two groups of individuals with VRFs: those without VRCs and those with VRCs receiving drug treatment for high BP, impaired fasting glucose (IFG), or abnormal lipid metabolism (ALM), each of which chronically results in the onset of HT, DM, or ALM, respectively. In the drug treatment group, the BHQs of WMH were compared according to the comorbidity of HT, DM, or DLP because these VRCs commonly combine together. Nonetheless, the relationship between WMH and comorbidity remains remarkably unclear [9,10,13]. To help make progress in this area, we designed and executed a large scale, cross-sectional study covering healthy and non-healthy states ranging from preclinical to chronic HT, DM, and DLP to examine whether WMH can be used as a healthcare quotient to maintain a healthy state or prevent the onset and development of VRCs.

## 2. Materials and Methods

### 2.1. Participants

Data were collected between January 2013 and April 2017 from the brain dock center (BDC) affiliated with Kochi University of Technology. From BDC, we enrolled 8921 healthy participants without a history of cerebral stroke, who underwent the brain dock health checkups only once. Although we were interested in the WMH of individuals with various medical backgrounds, participants who had been clinically diagnosed with HT, DM, and DLP but had not been treated with drugs were excluded (*n* = 1411). Thus, 5541 participants (2760 males, 2781 females; age, 20–89 years; mean age ± SD, 51.38 ± 9.80 years; median age, 51 years) with no medical history for HT, DM, and DLP were selected for analysis (Table 1). Here, the term “medical history” refers to the drug treatment history before and at the time of enrollment in the study. In addition, the following participants were also enrolled for analysis based on the examination results at BDC that compared WMH-BHQ of the participants with no medical history with that of those with a medical history of HT, DM, and DLP or their comorbidities (*n* = 1969).

All participants lived in Kochi Prefecture, visited BDC, and underwent brain MRI as part of their routine health checkups. They also answered a questionnaire on their past and present medical history and lifestyles, such as smoking. Health checkups included systolic blood pressure (SBP), body mass index (BMI), Brinkman index (BI; multiplying the average number of cigarettes smoked per day by the number of years the person has smoked), and various blood chemistry test items, including hemoglobin A1c (HbA1c), fasting blood sugar (FBS), triglycerides (TG), and high- density lipoprotein (HDL) and low-density lipoprotein (LDL) cholesterol. Based on these tests, BMI, BI, High BP, IFG, and ALM were classified according to the criteria shown in Table 2.

### 2.2. Automated Measurement of WMH Volume

A 1.5 Tesla MRI system (ECHELON Vega; Hitachi Medical Corporation, Tokyo, Japan) was used to perform MRI examinations for WMH diagnosis. The imaging protocol included T2-weighted spin-echo (repetition time/echo time (TR/TE) = 5800/96 ms), T1-weighted spin-echo (TR/TE = 520/14 ms), and fluid-attenuated inversion recovery (FLAIR; TR/TE = 8500/96 ms; inversion time = 2100 ms) images as described previously [16]. Images were obtained as 27 transaxial slices per scan. The slice thickness was 5 mm, with no interslice gap, as described previously [11,16]. Measurement of WMH volume was needed to evaluate the severity, especially for levels more than the maximum of the Fazekas scale. In our study, WMHs were automatically segmented and quantified for their volume using the following procedure. First, the FLAIR images were segmented into gray and white matter and cerebrospinal fluid space using SPM12, which also yielded the intensity inhomogeneity corrected image (IICI) [17,18]. Then, IICI was anatomically normalized into the template space using advanced normalization tools. A region-of-interest delineating the middle cerebellar peduncle was applied to the anatomically normalized IICI to estimate the intensity distribution of normal white matter of each subject. IICI in native space was then normalized for its intensity in the brain region segmented by the gray and white matter. The intensity normalized IICI was thresholded using a 3.5 SD cutoff to segment WMH, with search regions limited to WMH mask. The WMH volume (WMHV) was calculated by multiplying the voxel size by slice thickness. Finally, the measured WMH was automatically colored red to be detected by the first author (K.P.) who were a neurosurgeon trained enough to confirm the presence and location of WMH.

### 2.3. WMH-BHQ, a Novel Quotient Based on WMHV

Based on the WMHV of each participant, we constructed a new metric, the WMH-BHQ in that higher values are better and the median value for a given set of subjects is 100. In other words, this new metric was devised to convert WMHV to the standardized scale so that one can easily understand whether the WMHV of a subject is more or less than the median. In the development of this metric, we realized that the distribution of WMHV is skewed, and therefore, we used the percentile rank to define WMH-BHQ, which is the percentage of scores in its frequency distribution that are equal to or lower than it. For example, a test score that is greater than 75% of the scores of people taking the test is said to be at the 75th percentile, where 75 is the percentile rank. From the raw WMHV we obtained the percentile ranking for each subject (WMHV percentile), where zero means the lowest WMHV and 100 percentile means that the participant has the highest WMHV in the group. From this ranking method, a cumulative probability curve was estimated so that the percentile rank with newer data could be calculated with this curve. This estimation was based on the nonparametric density estimation and implemented by using the polspline function in the “polspline” package with R 3.4.3. We then defined this new brain health metric, WMH-BHQ, to be:WMH-BHQ = 100 + 15 × (50 − WMHV percentile)/24,(1)

With this formula, the median value, which was equivalent to the 50th percentile, generates a WMH-BHQ equal to 100. Likewise, the 74th and 26th percentiles produce BHQ 85 and 115, respectively. Originally, we considered directly using interquartile range (i.e., 75th and 25th percentiles). However, if the newer data were beyond the range of original data, an error would be produced. To avoid this prospect, we used the value 24 instead of 25 in the denominator. As a result, the WM-BHQ of 95% of our subjects ranged from 70.31 to 130. A lower WMH-BHQ means a higher and more problematic level of WMHV. A histogram of the WMH-BHQ values of our subjects with upward sloping curve implies “well-being” of brain health and a downward slope shows “not well-being” in terms of WMH.

### 2.4. Statistical Analysis

WMHV and WMH-BHQ data were not normally distributed. Thus, Mann–Whitney *U* and Kruskal–Wallis tests were utilized to evaluate the associations between WMHs and VRCs or other possible risk factors by comparing the differences between group distributions. The groups were defined by the presence or absence of VRCs or by standard criteria and classifications of risk factors described in the upper paragraphs. When the null hypotheses were rejected in the Kruskal–Wallis tests, we used Dunn tests [18] for pairwise comparisons. The *p* values were then adjusted using the Benjamini–Hochberg procedure [19], which controls the false discovery rate for multiple comparisons. Multiple regression analyses were performed to examine complex associations among multiple variables while controlling for the effect of potential confounding factors [20]. All statistical analyses except for the Benjamini–Hochberg procedure were performed using the Statistical Package for the Social Sciences software version 22 (IBM Corp., Armonk, NY, USA) [20]. Adjusted *p* values based on the Benjamini–Hochberg procedure were calculated using a Microsoft Excel (Microsoft Inc, Redmond, WA, USA) spreadsheet [21].

### 2.5. Standard Protocol Approvals, Registrations, and Participant Consents

Written informed consent was received from all participants and this study was reviewed and approved by the institutional review board of Kochi University of Technology.

### 2.6. Data Availability

Anonymized data might be shared by request.

## 3. Results

### 3.1. WMH-BHQ of Participants with no Medical History According to Age Decades

As shown in Figure 1a, the contour of the histogram for our metric WMH-BHQ changed remarkably across the different ages of the participants in this study. For subjects in their 40s, it was up right, while for those in their 50s, it was symmetric like a rainbow curve, and those in their 60s, 70s, and 80s generated plots shifting up and to the left as the age decades increased. In terms of WMH-BHQ, brain health obviously declined with age, as shown by the box plot in Figure 1b. The Kruskal–Wallis test and Dunn tests showed that all pairwise comparisons were significant except for those between 70s and 80s (*p* = 0.076). The WMHV histogram was asymptotic with a peak volume of <5 mL; therefore, WMH-BHQ was clearly superior to WMHV with regard to visualization and understanding of changes occurring over the age decades.

### 3.2. WMH-BHQ Histograms of Sex

There was a distinct difference in WMH-BHQ between males and females (Figure 2). The histogram of males was a trapezoid with an upward slope, while that of females showed a plateau, suggesting that females are more susceptive to WMH than males. 

Compared with results across the various age decades, the Mann–Whitney *U* test showed a clear sex difference for participants younger than their 50s but not for participants in their 60s and beyond (Table 3).

### 3.3. Analysis of WMH-BHQ Risk Factors: No VRCs

Multiple linear regression analysis of participants without VRCs was performed using age, sex, BMI, BI, SBP, HbA1c, FBS, TG, HDL, and LDL as independent variables and WMH-BHQ as a dependent variable (Table 4). A stepwise model was adapted for variable selection procedure. Female or male sex was a significant risk factor lowering or raising WMH-BHQ, respectively. The increases in age, SBP, and BI were significantly associated with the decrease in WMH-BHQ, whereas the increases in BMI, FBS, and TG were significantly associated with the increase in WMH-BHQ. HbA1c, HDL, and LDL were excluded after the stepwise regressions.

### 3.4. WMH-BHQ without VRCs: the Effect of Three Classifications and Two Criteria

We explored in detail the impact of differing values for BMI, BI, high BP and the criteria of IFG and ALM. Regarding BMI, the box plots in Figure 3a showed that WMH-BHQ significantly increased as the classification of BMI became larger. The Kruskal–Wallis test and the following Dunn tests showed all pairwise comparisons to be significantly different. BMI was also positively associated with WMH-BHQ. The effect of cigarette smoking, as measured by the BI, was that lower levels were associated with higher WMH-BHQ values. In particular, a BI value of 0 (Dunn test; *p* < 0.001) and 0–400 (*p* < 0.001) revealed significantly better bran brain health than levels above 400 (Figure 3b). Also, from Figure 4a, it is apparent that WMH-BHQ declined as SBP increased. Regarding high BP, WMH-BHQs of ≥400 (*p* < 0.010) and 0–400 (*p* < 0.001) revealed significant decreases compared with levels of ≥400 (Figure 4a). For IFG, a significant relationship existed only between (FBS < 100 and HbA1c < 5.6%) and (FBS ≥ 100 and <110 or HbA1c ≥ 5.6% and < 6.0%) (*p* < 0.001), although the other pair matches showed no significance (Figure 4b). TG showed a significant relationship only between 30 ≤ TG < 149 and 150 ≤ TG < 399 (*p* = 0.001), although the other pair matches showed no significance (Figure 4c). The LH ratio showed significant differences between 1 ≤ LH ratio < 1.5 and 2.5 ≤ LH ratio (*p* = 0.010) and between 1.5 ≤ LH ratio < 2.0 and 2.5 ≤ LH ratio (*p* = 0.018), although the other pair matches showed no significance (Figure 4d).

### 3.5. WMH-BHQ with VRCs and Their Comorbidities

WMH-BHQ histograms showed a downward slope for HT, while those of DM and DLP are almost plateaued compared with HT (Figure 5a). We also analyzed WMH-BHQ patterns regarding various multimorbidities, specifically, HT+DM, HT+DLP, DM+DLP, and HT+DM+DLP. Somewhat surprisingly, DM+DLP and HT+DM+DLP showed no downward slopes. Box plots showed that “no medical history” had the highest and HT+DM had the lowest median WMH-BHQs (Figure 5b). The Kruskal–Wallis test and the following Dunn test showed that all but DLP+DM had a significant difference in mean rank compared with no medical history (Table 5).

In addition, there was a statistically significant difference in the mean rank of WMH-BHQ between DLP and HT (*p* = 0.001), and a marginally significant difference between DLP and DM (*p* = 0.098). Furthermore, there were significant differences in the mean rank of WMH-BHQ between DLP and HT+DM (*p* < 0.001) and between HT+DM and DM+DLP (*p* = 0.022). These results show that DLP may positively affect WMH-BHQ, or at least, DLP appears unlikely to be a negative factor in this regard.

## 4. Discussion

Age was the strongest risk factor positively related to WMH on multiple regression analysis. In addition to age, SBP and BI were positive factors for WMH, which is consistent with prior reports [4,5]. Conversely, female sex, FBS, TG, and BMI were negative factors for WMH. The reason for sex differences in WMH remains unclear although the risk factors for LVD include male sex [22,23]. Recently, a population-based cognitively unimpaired cohort study with participants aged >70 years demonstrated that females had significantly greater WMHV than males [24]. Although our study examined the percentile of WMHV, the female susceptibility to WMH was significantly observed in participants aged <60 years but not in those aged >60 years (Table 3). The average menopausal age in Japan is around 50 years, but the differences across individuals are also regarded as substantial. Further study of the exact menopausal age is needed to elucidate the influence of sex hormones. If anything, sex differences according to age may contribute to the development of sex-specific preventive strategies against WMH progression linking to stroke and dementia.

Regarding fasting blood glucose and triglycerides, both β values were relatively low and the Kruskal–Wallis test showed no significant relationships according to IGF and ALM criteria. Fasting blood glucose and triglycerides may not be so heavily involved in WMH onset and development. Conversely, BMI was a powerful negative factor that increased the strength according to BMI criteria. BMI is an obesity index, whereas the others indicate waist circumstance (WC) and hip-waist ratio (HWR) [25]. Obesity directly depends on the amount of adipose tissue that can increase without weight gain. Obesity can be evaluated more accurately by WC or HWR than BMI, especially in the elderly [25]. Instead of BMI, WC was used for multiple regression analysis and yielded similar result as that with BMI, a negative factor of WMH. That obesity appears to suppress WMH may be because some growth factors for vessels are reportedly secreted from adipose tissue [26]. For example, angiopoietin-like protein 4 (ANGPTL4) is a member of the angiopoietin family, which encodes a secretory glycoprotein highly expressed in adipose tissue and liver and placenta [27]. ANGPTL4 and/or other vascular trophic factors are delivered from adipose tissue to the brain small vessels and may prevent WMH onset and progression. Further study will be needed to validate the hypothesis.

The existence and progression of WMH in the brain can be visually recognized through WMH-BHQ before or after VRCs such as HT, DM, and DLP emerge, that is, at any progression of morbidity from preclinical to chronic stages. The visible changes of WMH in the brain readily force patients to make efforts to control VRCs, linking to the prevention of stroke or dementia, and can determine when to start drug treatments or how to promote nondrug therapies, such as low calorie and salt diets and/or physical exercise. For example, HT diagnosis based on BP values remains unreflective of organ damages caused by a high BP. If WMH was accurately grasped by means of the new metric WMH-BHQ and efficiently treated according to the change in this measure, the damages due to HT in the brain and entire body could be minimized. Thus, WMH-BHQ may be regarded as an effective indicator that can help us to visualize brain damages and estimate the health of the whole body in terms of cerebral small vessel damages. Another advantage of WMH-BHQ is that it is based on percentile and ranking order, which enables minimization of measurement bias compared with WMHV, which heavily depends on scanning conditions and the abilities of MRI equipment.

Regarding comorbidity, HT+DM yielded lower WMH-BHQs than the other double morbidities (HT+DLP and DM+DLP) as well as triple morbidities (HT+DM+DLP), although this may be due to the possible ceiling effect of the WMH volume associated with the VRF. Multimorbidity is becoming a global challenge to prevent stroke and extend lifespan [28,29]. A nationally representative cross-sectional study of more than 1.4 million persons in Scotland showed that a diagnosis of stroke significantly became more common as the number of morbidities increased [30]. Additional comorbidities are widely considered to decrease health with a destructive metabolic domino effect [29]. In our study, however, DLP may have a suppressive or preventive effect on WMH in comorbidity, although the Scotland study did not describe DLP at all. According to the ALM criteria, there were no significant differences in WMH-BHQ. This evidence might imply some effect of statins, usual drugs for DLP, rather than DLP pathology. Several meta-analyses of placebo-controlled randomized trials suggest that statins may be beneficial in reducing the overall incidence of stroke [31,32]. It remains to be determined whether statins prevent onset and development of stroke through suppression of WMH. In our study, the numbers of DLP and DM patients were extremely smaller than those with HT. Further validation needs a larger number of participants with DLP and DM for the next follow-up study. In the near future, MRI parameters could be assessed through artificial intelligence pivoting on data mining [33,34]. Such approach together with the identification of biomarkers based on novel nanotechnology or biomedical engineering platforms would allow to propose new biosignatures for risk stratification in neurovascular patients [35].

### Limitations

The Brain Dock program utilizes an MRI-based approach to preventive medicine that was uniquely developed in Japan, aiming at early detection of unruptured cerebral aneurysms. At Brain Dock, health checks are conducted for a vast number of participants, and therefore, a large database could be built for brain research. Our study covered approximately 9000 participants living in Kochi Prefecture, Japan, and a single MRI machine was used throughout the study. Thus, the selection and information bias in this study could be minimized. However, our study included a bias of socioeconomic state involved, whereas one-fourth of the participants belonged to the white-collar class, such as public officials with moderate yearly incomes. The socioeconomic impact of this proportion is likely considered significant on WMH and the onset as well as the progression of VRCs to no small extent. The usage of 1.5 T MRI yields lower measurement of WMH volume as compared with 3 T MRI [36]. Nevertheless, the difference in magnetic power may be minimized in case of WMH-BHQ using the percentile of WMH volume. Our study was designed as a cross-sectional approach and only referred to the associations with WMHs at preclinical and chronic stages of VRCs. The next step will involve a prospective cohort study to certify the causal validation of WMH-BHQ using participants undergoing Brain Dock examinations more than twice.

## 5. Conclusions

In this study, we showed that cerebral white matter hyperintensities can be used as a healthcare quotient for quantitatively evaluating vascular risk factors or vascular risk conditions. Because of easiness of interpretation (Higher WMH-BHQ is better for brain in terms of cerebral vascular risk), WMH-BHQ might be useful for both clinicians and patients/inidividuals to pay their attentions to reduce cerebral vascular risks.

## Figures and Tables

**Figure 1 jcm-08-01823-f001:**
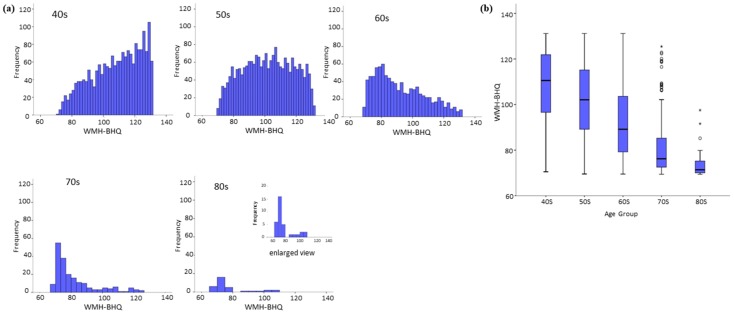
Histograms (**a**) and box plots (**b**) of WMH-BHQ with no medical history according to age decades of 40s, 50s, 60s, 70s, and 80s.

**Figure 2 jcm-08-01823-f002:**
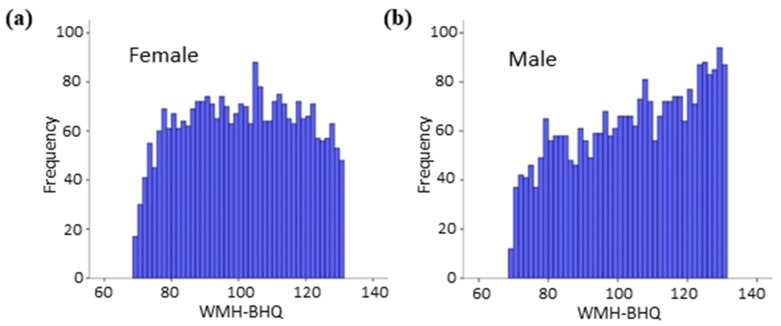
WMH-BHQ Histograms of females (**a**) and males (**b**).

**Figure 3 jcm-08-01823-f003:**
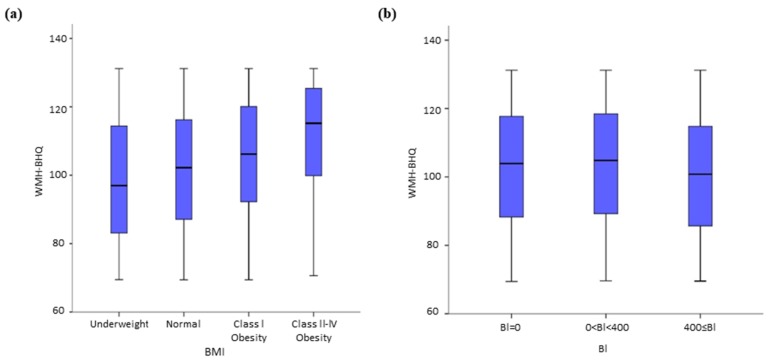
Box plots of WMH-BHQ with no medical history according to (**a**) Body Mass Index (BMI) and (**b**) Brinkman Index (BI).

**Figure 4 jcm-08-01823-f004:**
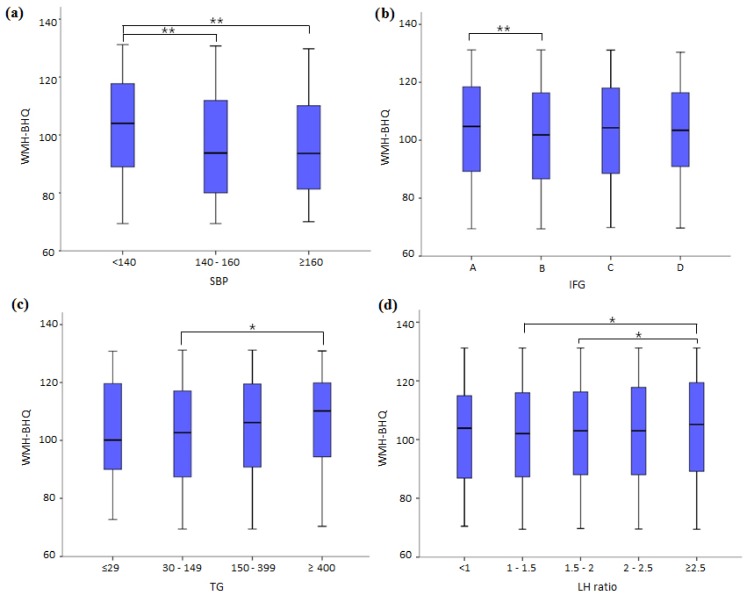
Box plots of WMH-BHQ with no medical history according to (**a**) systolic BP (SBP), (**b**) triglyceride (TG), (**c**) impaired fasting glucose (IFG) criteria, and d) the ratios of LDL to HDL (LH ratio). (**d**) A: fasting blood sugar (FBS) < 100 and HbA1c < 5.6; B: 100 ≤ FBS < 110 or 5.6 ≤ HbA1c < 6.0; C: 110 ≤ FBS < 126 or 6.0 ≤ HbA1c < 6.5; D: FBS ≥ 126 or HbA1c ≥ 6.5. * *p* < 0.05; ** *p* < 0.001

**Figure 5 jcm-08-01823-f005:**
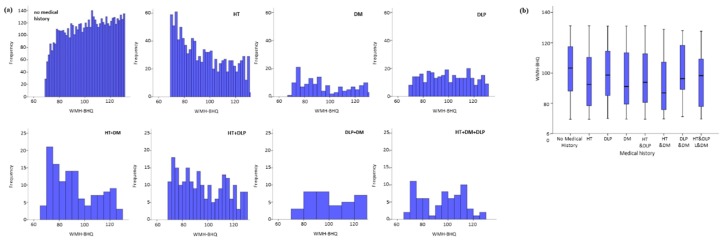
Histograms (**a**) and box plots (**b**) of WMH-BHQ according to no medical history, single morbidity, and multiple comorbidities. HT: Hypertension, DM: Diabetes mellitus, DLP: Dyslipidemia.

**Table 1 jcm-08-01823-t001:** Number and age distribution of participants without and with hypertension (HT), diabetes mellitus (DM), and/or dyslipidemia (DLP).

	Total	Male	Female	Mean Age ± SD (Years)	Median Age (Years)
No medical history	5541	2760	2781	51.4 ± 9.8	51
HT only	1074	622	422	59.5 ± 9.3	59
DM only	150	114	36	59.3 ± 9.8	59
DLP only	299	124	175	56.9 ± 8.7	57
HT + DLP	220	124	96	60.0 ± 9.0	59
HT + DM	124	97	27	60.8 ± 9.0	60
DM + DLP	35	19	16	57.0 ± 7.6	57
HT + DM + DLP	67	39	28	60.4 ± 7.6	59

**Table 2 jcm-08-01823-t002:** Classification of body mass index (BMI), Brinkman index (BI), high blood pressure (BP), and criteria of impaired fasting glucose (IFG) and abnormal lipid metabolism (ALM).

	Classification and Criteria
BMI	Underweight, BMI < 18.5; normal, 18.5 ≤ BMI < 25, overweight, 25 ≤ BMI< 30; and obese, BMI ≥ 30
BI	BI = 0, 0 < BI < 400, and BI ≥ 400
High BP	Systolic blood pressure (SBP) < 139, 140 ≤ SBP < 160, SBP ≥ 160
IFG	A (fasting blood sugar (FBS) < 100 and HbA1c < 5.6%) B (100 ≤ FBS < 100 or 5.6% ≤ HbA1c < 6.0%) C (110 ≤ FBS < 126 or 6.0% ≤ HbA1c < 6.5%) D (FBS ≥ 126 or HbA1c ≥ 6.5%)
ALM	Triglycerides (TG); TG < 29, 30 ≤ TG< 140, 140 ≤ TG< 400, and TG ≥ 400 Ratios of LDL to HDL (LH ratio): LH ratio < 1, 1 ≤ LH ratio < 1.5, 1.5 ≤ LH ratio < 2, 2 ≤ LH ratio < 2.5, and LH ratio ≥ 2.5

**Table 3 jcm-08-01823-t003:** Mann-Whitney U test of white matter hyperintensity brain healthcare quotient (WMH-BHQ) without medical history according to genders and age decades.

Mann-Whitney U (M-W U) Test							
	Male		Female				
Age Decade	N	Mean Rank	N	Mean Rank	M-W U	Z	P
30s	351	276.7	239	323.1	35361	−3.24	0.001
40s	955	855.0	865	971.8	359991	−4.74	0.001
50s	920	971.9	1117	1057.8	470485.5	−3.28	0.001
60s	437	449.5	460	448.5	100299.5	−0.05	0.957
70s	89	88.0	82	83.9	3473	−0.54	0.586
80s	8	14.9	18	12.9	61	−0.61	0.541

**Table 4 jcm-08-01823-t004:** Multiple linear regression analysis for white matter hyperintensity brain healthcare quotients (WMH-BHQ) risk factors.

Model	Unstandardized Coefficients		Standardized Coefficients	t	*p*	95% Confidence Interval for B	
	B	Standard Error	Beta			Lower Bound	Upper Bound
(Constant)	125.170	2.256		55.49	<0.001	120.750	129.595
Age	−0.064	0.022	−0.377	−29.53	<0.001	−0.708	−0.620
Body mass index (BMI)	0.693	0.070	0.135	9.86	<0.001	0.556	0.831
Systolic blood pressure (SBP)	−0.079	0.015	−0.074	−5.41	<0.001	−0.108	−0.051
Brinkman index (BI)	−0.003	0.001	−0.049	−3.88	<0.001	−0.004	−0.001
Fasting blood glucose (FBG)	0.045	0.015	0.038	2.96	0.003	0.015	0.074
Triglycerides (TG)	0.008	0.003	0.036	2.75	0.006	0.002	0.013

**Table 5 jcm-08-01823-t005:** Dunn test of WMH-BHQ among no medical history, hypertenstion (HT), diabetes mellitus (DM), dyslipidemia (DLP), and the comorbidities.

Pairwise Comparison by Dunn Test						
Comparison 1	Comparison 2	Test Statistic	SD	Z	Unadjusted *p*-Value	Adjusted *p*-Value
No medical history	HT	911.74	72.29	12.61	<0.001	<0.001
	DLP	368.32	128.72	2.86	0.004	0.015
	DM	793.29	179.40	4.42	<0.001	<0.001
	HT + DLP	765.91	149.05	5.14	<0.001	<0.001
	HT + DM	1307.35	196.87	6.64	<0.001	<0.001
	DLP + DM	188.32	367.63	0.51	0.608	0.710
	HT + DLP + DM	928.24	266.47	3.48	<0.001	0.002
HT	DLP	−543.42	141.77	−3.83	<0.001	0.001
	DM	−118.45	188.98	−0.63	0.531	0.676
	HT + DLP	−145.83	160.45	−0.91	0.363	0.485
	HT + DM	395.61	205.63	1.92	0.054	0.098
	DLP + DM	−723.42	372.40	−1.94	0.052	0.098
	HT + DLP + DM	16.49	273.01	0.06	0.952	0.952
DLP	DM	424.97	216.93	1.96	0.050	0.098
	HT + DLP	397.59	192.58	2.07	0.039	0.098
	HT + DM	939.03	231.58	4.06	<0.001	<0.001
	DLP + DM	−180.00	387.33	−0.47	0.642	0.719
	HT + DLP + DM	559.91	293.05	1.91	0056	0.098
DM	HT + DLP	−27.38	229.57	−0.12	0.905	0.939
	HT + DM	514.06	263.15	1.95	0.051	0.098
	DLP + DM	−604.97	406.99	−1.49	0.137	0.211
	HT + DLP + DM	134.94	318.59	0.42	0.672	0.724
HT + DLP	HT + DM	541.44	243.46	2.22	0.026	0.073
	DLP + DM	−577.59	394.55	−1.46	0.143	0.211
	HT + DLP + DM	162.32	302.53	0.54	0.592	0.710
HT+DM	DLP + DM	−1119.08	414.99	−2.70	0.007	0.022
	HT + DLP + DM	−379.12	328.74	−1.15	0.249	0.348
DLP+DM	HT + DLP + DM	739.91	452.18	1.64	0.102	0.168

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
