# Peer review of "Cerebral White Matter Hyperintensity as a Healthcare Quotient"

_jcm, 2019, doi:10.3390/jcm8111823_

Round 1
Reviewer 1 Report
The authors discussed the cerebral White Matter Hyperintensity, Leukoaraiosis, as a Healthcare Quotient to better understand the risk factors and optimal therapeutic strategies of cerebral white matter hyperintensity (WMH). This is an interesting topic and the authors are introducing the new concept of WMH-BHQ.
Minor comments:
The title should include either WMH or leukoaraiosis because they are referring to the same thing. In the introduction, line 95 the authors mentioned that the role of the DM and DLP is still disputable. It would be great if you can elaborate a little bit in this regard as the finding from this study needs such elaboration. Line 111, please define middle age and older age as this is not standard. Line 186, the 1.5 tesla should be mentioned in the limitation section. Line 193, the slice thickness is 5 mm but there should be an inter slice gap. If you have a reference saying that the MRI scan used in that study had no inter slice gap then please refer to that reference. Line 300 How TG and BMI is related to the WMH-BHQ? is increase BMI is associated with increase WMH-BHQ, or the opposite? Same question regarding the TG levels. Please clarify this point. Line 317-323, Is there a better way to explain this because it is confusing to the reader. You may want to talk about the positive results in brief here and refer to the table or figure for more details. Line 363, the authors mentioned MWH, Do you mean WMH?Lines 397-399, the authors mentioned "397
Regarding comorbidity, HT+DM yielded higher WMH-BHQs than double
morbidities (HT+DLP and DM+DLP) as well as triple morbidities (HT+DM+DLP). This may be due to the possible ceiling effect of the WMH volume associated with the VRF. Would you please elaborate on that. Also, please refer to the attached file with the highlighted paragraphs that cab be rephrased in a better way.
Author Response
To reviewer #1
The title of our revised manuscript was changed to “Cerebral White Matter Hyperintensity as a Healthcare Quotient”.
Line111; we defined middle age and older age with age range.
Line 186; the limitation of the 1.5 tesla was described in the Discussion.
Line 186; the 5mm slice thickness without inter slice gap was adopted in our study and two manuscripts were referred for this method.
Line 300; the increases or decreases in parameters were clarified according to WMH-BHQ.
Line 317-323; we explained A and B in Figure 4b in detail.
Line 363; we changed MWH to WMH.
Line 397-399; the possible ceiling effect was described as another explanation that HT+DM yielded lower WMH-BHQs than double morbidities.
Reviewer 2 Report
The work proposes a new healthcare quotient for classification of cerebral white matter hyperintensity.
It is interesting and pertinent. Please find below additional comments:
* line 169-170: "All participants...": remove sentence since mentioned in later paragraph
* section Methods/Participants: did the authors ensure that patients did not show up at Brain Dock several
times, e.g. because of a renewed control check-up ? If it is ensured that no such doubled
occurrences happend, please mention it.
* section from lines 173-184 repeats the content of the Table 2. Please remove.
* line 208: what is LA ?
* line 209: what is (K,P) ?
* line 222: WMNV --> WMHV
* line 249: "in Methods": the line is written in the Methods section itself, the authors may
replace "Methods" by "the upper paragraphs".
* line 309-310: "...all pairwise comparisons to be significant" : significant different ?
* enlarge all labels of all axis in all figures.
* Caption of Figure 1: mention what the numbers in figure panels mean (age decades).
Author Response
To reviewer #2
line 169-170; “ All participants…” was removed.
We added the explanation that participants underwent the Brain Dock health checkups only once.
line 173-184; These parts were totally removed.
line 208; We changed LA to WMH.
line 209; K.P. is the first author’s initial. We explained it.
line 222; We changed WMNV to WMHV.
line 249; We changed “Methods” to “the upper paragraphs”.
line 309-310; We changed “significant” to “significantly different”. We enlarged all labels of all axis in all figures.
Ade decades were explained in the caption of Figure1.
Reviewer 3 Report
The authors propose the use of MRI based white matter assessment as a predictive tool for neurovascular ischaemic pathologies. I enjoyed reading the article and would ask only one additional comment to be added in the discussion session:
In the near future, MRI parameters could be assessed through artificial intelligence pivoting on data mining (Garg R et al., 2019; Schirmer MD, et al., 2019) Such approach together with the identification of biomarkers based on novel nanotechnology or biomedical engineering platforms would allow to propose new biosignatures for risk stratification in neurovascular patients (Ganau L et al., 2018).
References:
Garg R, et al. Automating Ischemic Stroke Subtype Classification using machine learning and natural language processing. J Stroke Cerebrovasc Dis. 2019
Schirmer MD, et al. White matter hyperintensity quantification in large scale clinical acute ischemic stroke cohort. The MRI-GENIE study. Neuroimage Clin. 2019 Ganau L, et al. Understanding the mmMlinpathological basis of neurological diseases through diagnostic platforms based on innovations in biomedical engineering: new concepts and theranostic perspectives. Medicines (Basel). 2018
Author Response
To reviewer #3
The comment with three references was added in Discussion.